# Caveolin-1 Down-Regulation Reduces VEGF-A Secretion Induced by IGF-1 in ARPE-19 Cells

**DOI:** 10.3390/life12010044

**Published:** 2021-12-28

**Authors:** Alessandra Puddu, Roberta Sanguineti, Davide Maggi

**Affiliations:** Department of Internal Medicine and Medical Specialties, University of Genova, Viale Benedetto XV, 6, 16132 Genova, Italy; roberta.sanguineti@gmail.com (R.S.); davide.maggi@unige.it (D.M.)

**Keywords:** caveolin-1, vascular endothelial growth factor A, insulin-like growth factor-1, ocular neovascularization, retinal pigment epithelium, endothelial cells

## Abstract

The insulin-like growth factor 1 (IGF-1) stimulates expression and secretion of vascular endothelial growth factor-A (VEGF-A), the main actor in ocular neovascularization, by RPE cells. Activity of IGF-1 is regulated by interaction between its receptor and Caveolin-1 (Cav-1), the main component of caveolae. The aim of this study was to investigate whether modulation of Cav-1 expression affects synthesis and secretion of VEGF-A. ARPE-19 cells were transfected with small interfering RNA for Cav-1 (si-Cav-1) and with control siRNA (si-CTR) and stimulated with IGF-1. We found that down-regulation of Cav-1 did not affect activation of IGF-1R but regulated in an opposite manner the phosphorylation of Akt and Erk1/2. Moreover, we found that IGF-1 increased mRNA levels of VEGF-A in both si-CTR and in si-Cav-1 ARPE-19 cells and that Cav-1 silencing significantly reduced basal and IGF-1-stimulated VEGF-A release. Then we investigated the response of the microvascular endothelial cell line HMEC-1 to secretory products of ARPE-19 cells by evaluating wound healing closure, finding that conditioned media from si-Cav-1-ARPE-19 cells reduced endothelial cell migration rate. These data demonstrate that Cav-1 regulates secretion of VEGF-A, and that the depletion of Cav-1 reduces IGF-1 induced VEGF-A secretion in ARPE-19 cells and the migratory potential of their secretory products.

## 1. Introduction

Caveolin-1 (Cav-1) is the principal protein of caveolae, specialized lipid rafts which regulate many cellular functions, including endocytosis, intracellular trafficking and signaling [1]. Caveolae are broadly diffused and are described also in the retinal pigment epithelium (RPE), a monolayer of highly specialized cells located between the retinal photoreceptors and the choroidal vasculature [2]. RPE cells play multiple roles in the retina: they produce several growth factors, are part of the outer blood-retinal barrier (BRB), perform transcytosis and phagocytosis, and digest the photoreceptor outer segments [2,3,4]. Although Cav_−/−_ mice develop a normal RPE [5], caveolin-1 seems to play an important role in RPE function. Indeed, it affects ion transport activities and is required for phagolysosomal digestion of photoreceptors [5,6]. RPE cells are a main source of vascular endothelial growth factor-A (VEGF-A), which plays a critical role in maintaining the homeostasis of the retinal and choroidal vasculature [3]. It has been well established that VEGF-A also sustains pathological neovascularization of the retina, leading to blinding pathologies such as wet age-related macular degeneration (AMD), proliferative vitreoretinopathy (PVR), and diabetic retinopathy. Studies on the role of Cav-1 in ocular neovascularization have led to controversial results. Indeed, caveolin-1 siRNA inhibition reduced retinal neovascularization in a murine model of oxygen-induced retinopathy [7] but worsened choroidal and retinal neovascularization in Cav-1-deficient mice [8].

Evidence has shown that insulin-like growth factor-1 (IGF-1) may act as a pro-angiogenic agent in the eye, not only for mitogenic effect on endothelial cells, but also by stimulating VEGF-A secretion by RPE cells [9,10,11]. It is well known that binding of IGF-1 to its receptor (IGF-1R) results in receptor auto-phosphorylation, leading to the activation of several intracellular signaling, including the PI3K/Akt and MAPK pathways [12]. Since we previously demonstrated that IGF-1R localizes in caveolae and that IGF-1 signaling may be regulated by Cav-1 [13,14], the aim of this study was to investigate whether Cav-1 is involved in mediating IGF-1-induced VEGF-A expression and secretion in the RPE cell line ARPE-19.

## 2. Materials and Methods

### 2.1. Cell Culture

The human cell line ARPE-19 passage 22 to 28 (American Type Culture Collection, Manassas, VA, USA) were cultured in DMEM/F12 (Cambrex Bio Science, Walkersville, MD, USA) supplemented with 10% FBS, antibiotics (100 U/mL penicillin G and 100 μg/mL streptomycin sulphate) and 2 mmol/L-glutamine (both from Sigma-Aldrich, Milan, Italy). HMEC-1 cells passage 23 to 26 (American Type Culture Collection, Manassas, VA, USA) were grown in MCDB131 medium supplemented with 10% FBS, 10 ng/mL epidermal growth factor, 1 μg/mL hydrocortisone, and 10 mM glutamine. Cells were maintained in a humidified 5% CO_2_ air incubator at 37 °C. The cell medium was replaced every 2 days. Once cells reached confluence, they were trypsinized, and seeded in multiwell plates.

### 2.2. RNA Silencing

For the Cav-1 siRNA experiment we used a pool of 3 prevalidated siRNA-targeting human caveolin-1 and 1 iBONi siRNA negative control (Riboxx GmbH, Radebeul, Germany). ARPE-19 cells were split into 12-well plates (5 × 10^4^ cells/well) and cultured overnight to reach 40–50% confluence. Then, cells were transfected with riboxxFECT mix (Riboxx GmbH, Radebeul, Germany) containing either Cav-1 siRNA (si-Cav-1) or CTR siRNA (si-CTR). Transfection mixtures were prepared according to manufacturer’s instruction and left on the cells for 72 h. Cav-1 protein expression level was evaluated by immunoblotting.

### 2.3. Immunoblotting Analysis

ARPE-19 cells were lysed in RIPA buffer supplemented with protease and phosphatase inhibitors (Pierce, Rockford, MD, USA) and the protein content was quantified using the BCA Protein Assay Kit (Pierce, Rockford, MD, USA) according to the manufacturer’s instructions. Thirty micrograms of total cell lysate were run on an SDS-PAGE and transferred onto nitrocellulose. Membranes were placed in the blocking solution (5% nonfat dried milk) and incubated overnight at 4 °C with primary specific antibodies diluted at 1:1000: anti-Akt (cat. n. 9272), anti-phospho-Akt (Ser473, cat. n. 4060), anti-caveolin-1 (cat. n. 3267), anti-IGF1R (cat. n. 9750), anti-phospho-IGF-1R (Tyr980, cat. n. 4568), anti-phospho-p44/42 MAPK (Erk1/2) (Thr202/Tyr204, cat. n. 9101), anti p44/42 MAPK (Erk1/2, cat. n. 9102), and anti-β-actin (1:3000, cat. n. 3700), from Cell Signaling Technology, Beverly, MA, USA. Membranes were incubated with horseradish peroxidase conjugated secondary antibodies for 1 h at room temperature. Bound antibodies were detected using the enhanced chemiluminescence lighting system (LiteAblot EXTEND, EuroClone, Milan, Italy), according to the manufacturer’s instructions. Each membrane was stripped (Restore PLUS Western Blot Stripping Buffer, Pierce Biotechnology, Rockford, IL, USA) and probed for β-actin to verify equal protein loading. Membranes immunoblotted with phIGF-1R, phAkt and phERK1/2 were stripped and reprobed respectively with anti-IGF-1R, anti-Akt and anti-ERK1/2 antibodies to normalize the blots for the total protein levels. Densitometric analysis of the bands of interest (p 42 has been used for ERK1/2 quantification) was performed using the Alliance software. Results were expressed as percentages of CTR (defined as 100%).

### 2.4. RNA Isolation, cDNA and qRT-PCR

RNeasy kit (QIAGEN s.r.l., Milan, Italy) was used to isolate total RNA from confluent ARPE-19 cells according to manufacturer’s instruction. Reverse transcription of one microgram of RNA to cDNA was performed using Wonder RT-cDNA Synthesis kit (Euroclone, Milan, Italy). qRT-PCR amplification of VEGF-A (Applied Biosystems assay ID: Hs00900055_m1) was performed using Luna Universal Probe qPCR Master Mix (New England Biolabs, NEB, Ipswich, MA, USA) in an ABI PRISM 7900 HT Fast Real Time PCR System (Applied Biosystems, Monza, Italy). Quantification was performed in triplicate with the following cycling conditions: initial denaturation (95 °C 1 min), followed by 43 cycles of denaturation (95 °C 15 s) and extension (60 °C 30 s). Gene expression level was normalized using β-Actin (Applied Biosystems assay ID: Hs01060665_g1) as control gene. The 2^−ΔΔCt^ method has been used to analyze changes in gene expression.

### 2.5. VEGF-A Secretion

Confluent ARPE-19 cells (wild type and transfected) were serum starved overnight before treatment for 4 h with 100 nmol/L recombinant human IGF-1 (Gibco by Life Technologies Corporation, Carlsbad, CA, USA) in serum-free medium. Then, the conditioned media were collected and stored at −80 °C until the assay to quantify VEGF-A secretion was performed. Quantitative secretion of VEGF-A was evaluated by ELISA (Bender MedSystem, Vienna, Austria). Concentration of VEGF-A was calculated from standards curve and each value was related to total protein concentration of the respective lysate.

### 2.6. Scratch Assay

HMEC-1 cells (5 × 10^4^ cells/well) were seeded in a 24-well plate. Once they reached confluence the cell layers were scratched with a pipette tip, and cells were washed to remove detached cells. Conditioned media collected from ARPE-19 cells (wild type and transfected) treated with or w/o 100 nmol/L recombinant human IGF-1 for 24 h were added to the wounds. Images of the damaged areas were taken at 4× magnification using a microscopic bright field immediately after the scratches and after 24 h of incubation. Quantification of cell migration rate was based on distance between the edges of the scratch [15].

### 2.7. Cell Viability

HMEC-1 cells were plated in a 96-well plate let to adhere, and then incubated for 24 h with conditioned media of ARPE-19 cells (wt and transfected) treated with or w/o 100 nmol/L recombinant human IGF-1. Cell Titer 96 Aqueous One Solution Cell Proliferation Assay (Promega, Milan, Italy) was used to detect viable cells according to the manufacturer’s instructions.

### 2.8. Statistical Analysis

Statistical analysis was performed with GraphPad Prism 4.0 software (GraphPad Software, San Diego, CA, USA) using one-way ANOVA followed by Bonferroni’s multiple comparison test. The results come from at least three experiments and were represented as the mean ± SD. A *p* value < 0.05 was considered as statistically significant.

## 3. Results

### 3.1. Caveolin-1 Expression and Down-Regulation of Cav-1 with siRNA

ARPE-19 cells constitutively express Cav-1. Silencing Cav-1 (si-Cav-1) by RNA interference resulted in ~80% reduction of Cav-1 expression compared with ARPE-19 cells transfected with scrambled siRNA (si-CTR) (Figure 1a). The amount of β-actin was used as internal control (Figure 1 and Appendix A).

All the results obtained in si-CTR ARPE-19 cells are comparable with those obtained in untransfected cells (CTR, CTR + IGF-1). However, we showed results from wild type (wt) cells for completeness.

### 3.2. Effects of Caveolin-1 Down-Regulation on Intracellular Signaling of IGF-1

Firstly, we verified that Cav-1 depletion did not affect expression of IGF-1R, Akt and Erk1/2. As shown in Figure 1b–d, expression of these proteins was not altered by Cav-1 down-regulation. It is well known that binding of IGF-1 to the IGF-1R results in receptor auto-phosphorylation (12), thus we investigated whether Cav-1 down-regulation affected activation of IGF-1R by IGF-1 (see Appendix A). Down-regulation of Cav-1 did not alter the ability of IGF-1 to stimulate autophosphorylation of IGF-1R (Figure 2a). Then we evaluated the ability of IGF-1R to phosphorylate its downstream targets Akt and Erk1/2. In si-Cav-1 ARPE-19 cells, phosphorylation of Akt in response to IGF-1 was significantly increased in comparison with si-CTR cells (Figure 2b). In contrast, activation of Erk1/2 by IGF-1 was significantly reduced in si-Cav-1 ARPE-19 cells compared with si-CTR cells (Figure 2c).

### 3.3. Effects of Caveolin-1 Down-Regulation on VEGF-A Expression and Secretion

To determine whether Cav-1 has a role in IGF-1 biological action, we evaluated mRNA expression and secretion of VEGF-A in ARPE-19 cells after Cav-1 down-regulation.

Treatment with IGF-1 significantly increased mRNA levels of VEGF-A in both si-CTR and si-Cav-1 cells (Figure 3a).

Under unstimulated conditions mRNA expression of VEGF-A was unaffected by Cav-1 depletion (Figure 3a), but the rate of VEGF-A secretion was reduced (Figure 3b). Stimulation with IGF-1 significantly increased VEGF-A secretion by almost two-fold compared with the basal value (Figure 3b). However, Cav-1 silencing reduced the ability of IGF-1 to rise VEGF-A secretion compared with the control siRNA transfected cells.

### 3.4. Effects of ARPE-19 Secretome on Endothelial Cell Activation

To investigate whether secretory products from ARPE-19 cells depleted of Cav-1 may differently affect the response of endothelial cells, we exposed HMEC-1 cells to the conditioned media collected from ARPE-19 cells (wt and transfected) and investigated the ability of HMEC-1 cells to repair damage using the scratch assay. Scratched confluent cell layers were cultured for 24 h in media collected from ARPE-19 cells treated with or w/o 100 nmol/L recombinant human IGF-1. The repair of the damaged area was comparable in HMEC-1 cells incubated with media from unstimulated ARPE-19 cells (Figure 4a). HMEC-1 cells cultured with media from si-Cav-1 + IGF-1 ARPE-19 cells displayed reduced cell migration rate and wound closure (Figure 4b).

To assess whether wound closure had been affected by any toxic effect, we evaluated cell viability of HMEC-1 cells cultured for 24 h with conditioned media of ARPE-19 cells treated with or w/o 100 nmol recombinant human IGF-1. No difference was found in cell viability among all culture conditions (Figure 4c).

## 4. Discussion

The primary aim of our work was to investigate whether depletion of Cav-1 may affect expression and secretion of VEGF-A induced by IGF-1 in ARPE-19 cells.

Firstly, we found that down-regulation of Cav-1 expression in ARPE-19 cells significantly reduced constitutive VEGF-A secretion. Evidence that caveolin-1 affects the release of several factors has been reported in different cell types. Serum levels of high-molecular-weight adiponectin were markedly reduced in Cav-1 null mice compared with wild type controls [16]. In pancreatic cell lines, knockdown of Cav-1 significantly increased insulin secretion under physiological glucose levels [17]. Therefore, the loss of Cav-1 and, consequently, the disruption of caveolae, may be connected to the impaired secretion of VEGF-A in Cav-1-depleted ARPE-19 cells, suggesting that Cav-1 has a role in regulating VEGF-A secretion. This hypothesis is supported by the finding that Cav-1 depletion did not affect the mRNA expression of VEGF-A in these cells, thus excluding the idea that the defective secretion is due to a decreased VEGF-A gene transcription.

Synthesis and secretion of VEGF-A may be stimulated by IGF-1 in RPE cells [9]. Levels of IGF-1 have been reported to be increased in the serum and vitreous of patients with proliferative diabetic retinopathy [18,19]. As expected, we found that treatment of ARPE-19 cells with IGF-1 significantly increased both expression and secretion of VEGF-A. In addition, we showed that depletion of Cav-1 in ARPE-19 cells prevented the rise of VEGF-A secretion induced by IGF-1. Once again, we found that the defect in VEGF-A release was not coupled to decreased levels of VEGF-A mRNA, strengthening the hypothesis that Cav-1 has a role in regulating VEGF-A secretion. The latest result is of particular importance, because the IGF-1-dependent signaling has been causatively related to the development of pathological ocular neovascularization due to VEGF stimulation [9,20].

Signal transduction pathway activated by IGF-1 is mediated by IGF-1R, which is located in caveolae [13]. We have previously shown that IGF-1R directly interacts with Cav-1 [14], and that Cav-1 down-regulation inhibits IGF-1 receptor signal transduction in rat cardiomyoblasts [21]. Based on this evidence, it is reasonable to hypothesize that depletion of Cav-1 may affect IGF-1 signaling and the consequent induced secretion of VEGF-A in RPE cells. However, we found that IGF-1R phosphorylation is not affected by down-regulation of Cav-1 in ARPE-19 cells. Therefore, the impairment of VEGF-A secretion is not related to a decreased activation of IGF-1R, suggesting that it may lie on its intracellular substrates. Indeed, its main downstream effectors, Akt and Erk1/2, were modulated in an opposite manner by Cav-1 down-regulation. This different molecular consequence of ligand activation of IGF-1R by IGF-1 may be related to the multiple roles of RPE cells in the retina. Activated Akt mediates a lot of cell signaling events, including proliferation and survival [22]. Interestingly, we found that phosphorylation of Akt is further up-regulated by IGF-1 in Cav-1 depleted ARPE-19 cells. It is well known that IGF-1 preserves RPE cells from damage via the activation of the PI3K/Akt signaling pathway [12,23,24]. Therefore, our results suggest that depletion of Cav-1 may improve the protective effects of IGF-1 on RPE cells. Moreover, we found that that IGF-1-induced Erk phosphorylation is significantly decreased in ARPE-19 cells in which Cav-1 has been depleted. Since it has been reported that the MAP Kinase pathways controls VEGF-A stimulated secretion in RPE cells [10,25,26], Cav-1 depletion may prevent IGF-1-induced VEGF-A secretion by attenuating the Erk kinase signaling.

RPE cells play an important role in maintaining retinal homeostasis. In particular, proteins secreted by RPE cells, such as VEGF-A, may affect the biological function of the neighboring cells, including the endothelial cells [27]. Secretory products are altered in several eye diseases [27]. Here we found that depletion of Cav-1 affected the composition of the ARPE-19 cells secretome by reducing levels of VEGF-A in response to IGF-1. Moreover, our results show accelerated wound healing in endothelial cells exposed to media from CTR -esi-CTR-ARPE-19 cells stimulated with IGF-1, but not in those exposed to media from si-Cav-1 ARPE-19, confirming that the migratory potential of the surnatant has been reduced by Cav-1 depletion. Finally, the lack of difference in proliferation of HMEC-1 cells support the hypothesis that media from si-Cav-1 ARPE-19 exposed to IGF-1 induced a lowered migration response in endothelial cells.

Some evidence highlights the important role of caveolae in vision-related functions [28]. Endogenous Caveolin-2 is almost undetectable, suggesting that Cav-1 is the main component of caveolae in RPE [29]. The role of Cav-1 in RPE cells is still controversial. Indeed, on one hand, cav-1 has been found overexpressed in PVR and choroidal neovascularization, suggesting that it is involved in promoting pathological ocular neovascularization [28]. On the other hand, other studies have shown that knockdown of Cav-1 enhanced epithelial to mesenchymal transition and increased blood retinal barrier permeability, key events for the onset of PVR [30,31]. Here we found that Cav-1 depletion decreased constitutive secretion of VEGF-A. It is well known that basal level of VEGF-A secreted by RPE cells is important in maintaining retina homeostasis [4], therefore, the decreased secretion of VEGF-A could affect retinal function and promote the onset of a microenvironment that leads to blood retinal barrier breakdown. On the contrary, increased secretion of VEGF-A is causally involved in pathological retinal neovascularization. In this case, downregulating VEGF-A secretion through Cav-1 depletion may be useful in counteracting excessive proangiogenic stimulation. Furthermore, increased levels of VEGF-A in the retina threaten the integrity of the blood retinal barrier by inducing transcytosis in endothelial cells [31]. All these considerations support the important role of caveolin-1 in maintaining VEGF-A homeostasis. Our study showed that caveolin-1 may play an important role in regulating both basal and IGF-1 induced VEGF-A secretion in RPE cells. Moreover, we demonstrated that caveolin-1 depletion in RPE cells decreases the migratory potential of their secretory products, suggesting that reducing expression of caveolin-1 may be useful in counteracting ocular neovascularization.

In conclusion, our results suggest that caveolin-1 may be a novel target in regulating VEGF-A homeostasis. Therefore, employment of strategies that modulate caveolin-1 expression may be taken into account in order to design therapeutic approach also in ocular neovascularization.

## Figures and Tables

**Figure 1 life-12-00044-f001:**
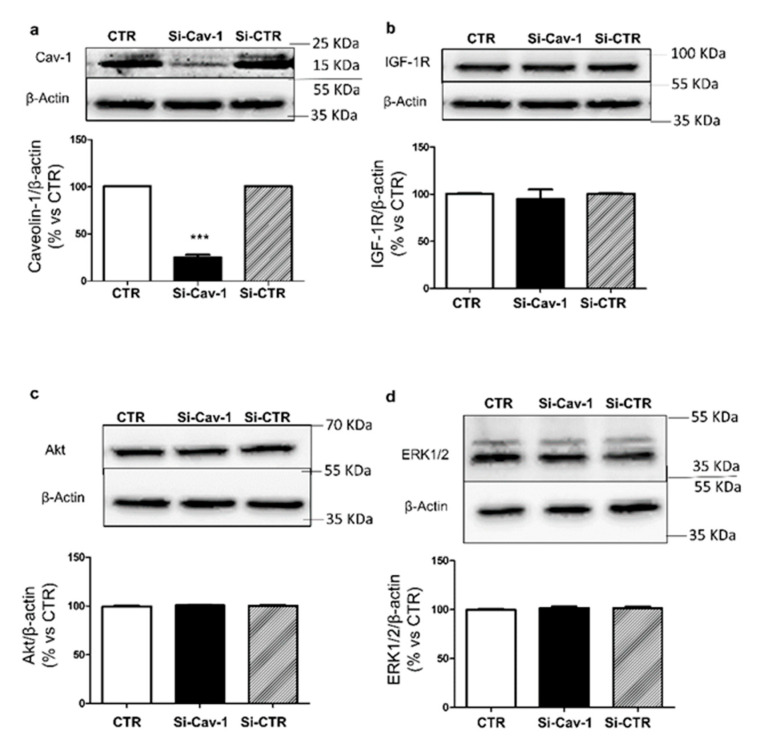
Cav-1 down-regulation in ARPE-19 cells. ARPE-19 cells were transfected with specific siRNA for Cav-1 (si-Cav-1) or a random sequence (si-CTR). Seventy-two hours from the transfection cells were lysed and immunoblotted with (**a**) anti-Cav-1, (**b**) anti-IGF-IR, (**c**) anti-Akt and (**d**) anti Erk1/2 and β-actin antibodies. Data are expressed as mean ± SD of fold induction relative to β-Actin. Blots are representative of three independent experiments. Each bar represents the mean ± SD (*n* = 3) *** *p* < 0.001 vs. CTR and si-CTR.

**Figure 2 life-12-00044-f002:**
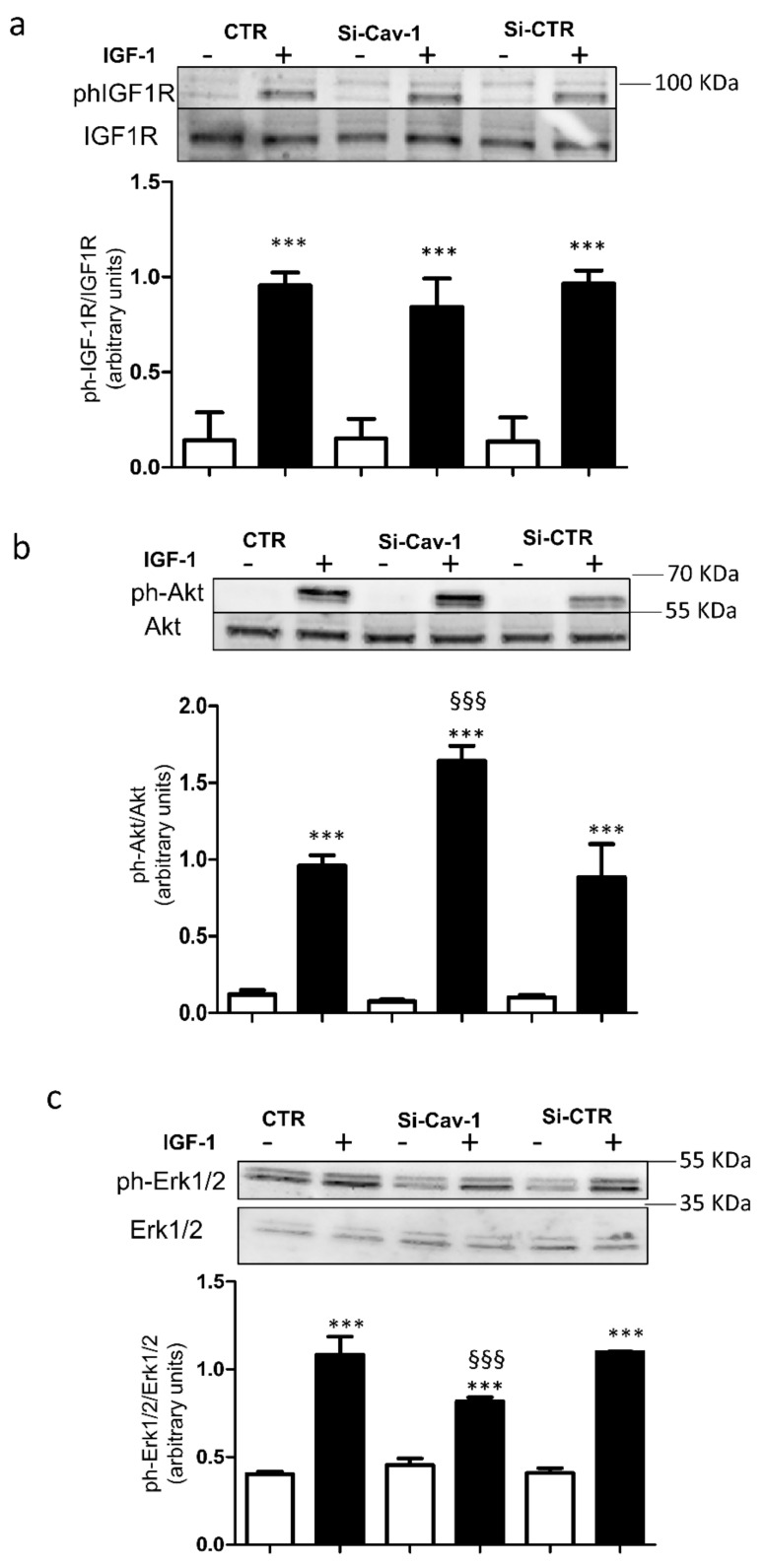
Effects of caveolin-1 depletion on intracellular signaling. ARPE-19 cells were transfected with Cav-1-siRNA or CTR-siRNA. After 72 h a batch of cells were serum starved overnight, then exposed in the presence (dark bars) or absence (white bars) of 100 nmol/L IGF-1 for 5 min. Then cells were lysed and immunoblotted with anti-phospho and anti-pan antibodies against IGF1R (**a**), AKT (**b**), and ERK1/2 (**c**). A representative Western blotting and quantification of densitometries of Western blot bands are shown. Data are expressed as mean ± SD of fold induction relative to β-Actin (*n* = 3). *** *p* < 0.001 vs. the respective unstimulated batches of cells; ^§§§^
*p* < 0.001 vs. IGF-1 and si-CTR + IGF-1.

**Figure 3 life-12-00044-f003:**
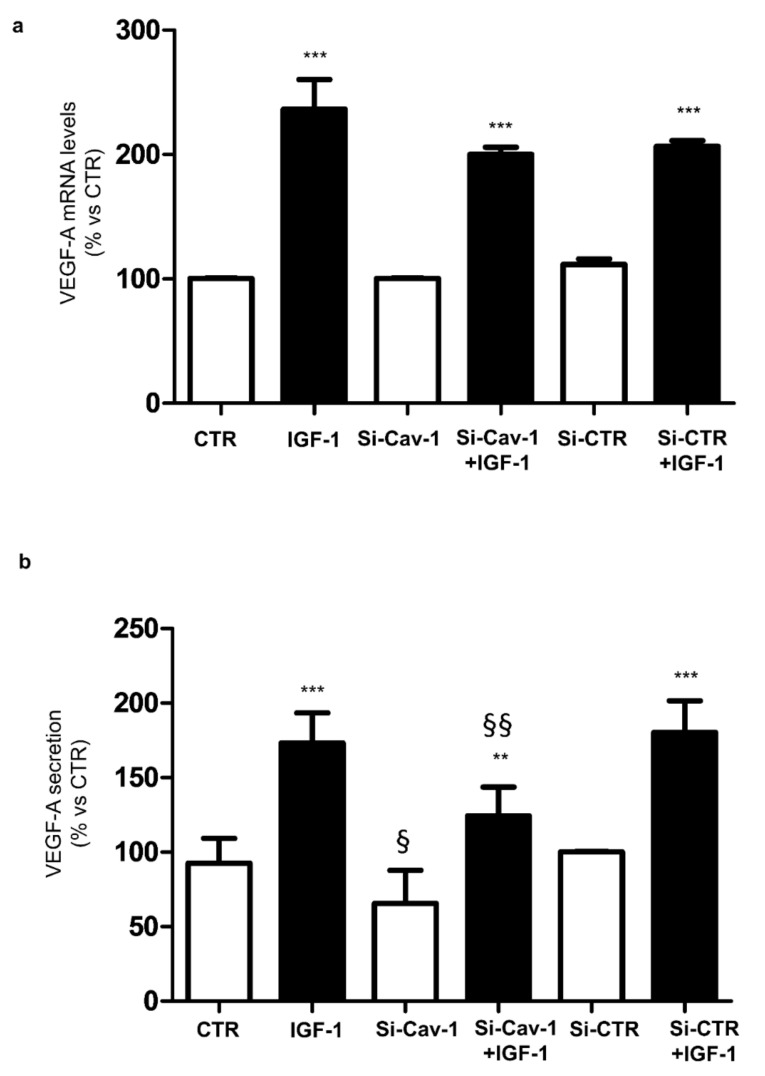
Cav-1 silencing decreased basal and IGF-1-induced VEGF-A secretion. ARPE-19 cells were transfected with Cav-1-siRNA or CTR-siRNA. After 72 h cells were serum starved overnight, then treated for 4 h with 100 nmol/L IGF-1. (**a**) Real-time PCR was performed using specific primers for VEGF-A. Gene expression was normalized using the housekeeping β-Actin as control gene. Each bar represents the mean ± SD (*n* = 3) *** *p* < 0.001 vs. the respective unstimulated batches of cells. (**b**) VEGF-A content into the collected buffers was assayed by ELISA. Data were normalized to cellular protein content. Each bar represents the mean ± SD (*n* = 3) ** *p* < 0.01 and *** *p* < 0.001 vs. the respective unstimulated batches of cells; ^§^
*p* < 0.05 vs. CTR and si-CTR; ^§§^
*p* < 0.01 vs. IGF-1 and si-CTR +IGF-1.

**Figure 4 life-12-00044-f004:**
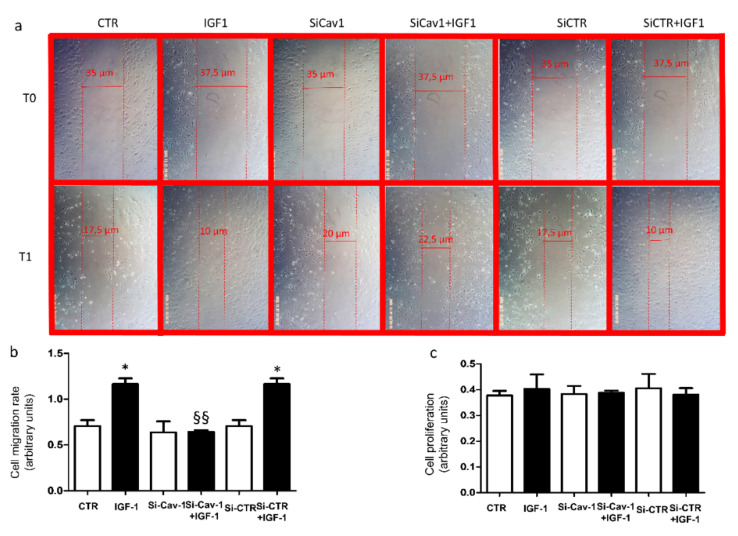
Scratch assay. (**a**) Representative images of the damaged areas immediately after the scratches (T0) and after 24 h (T1) incubation with conditioned media collected from ARPE-19 cells (wild type and transfected) treated with or w/o 100 nmol/L recombinant human IGF-1. (**b**) Quantification of cell migration rate from each of the six experimental cell groups [15]. Data are expressed as the mean ± SD of three independent experiments. * *p* < 0.05 vs. HMEC-1 cells exposed to respective conditional media collected from unstimulated ARPE-19 cells, as well as ^§§^
*p* < 0.01 vs. IGF-1 and si-CTR+IGF-1. (**c**) HMEC-1 cell viability. Cells were cultured for 24 h with conditioned media collected from ARPE-19 cells (wild type and transfected) treated with or w/o 100 nmol/L recombinant human IGF-1. Data are expressed as the mean ± SD of three independent experiments.

## Data Availability

The data presented in this study are available in supplementary materials, any other data are available on request from the corresponding author.

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
