# Peer review of "Caveolin-1 Down-Regulation Reduces VEGF-A Secretion Induced by IGF-1 in ARPE-19 Cells"

_life, 2021, doi:10.3390/life12010044_

Round 1

Reviewer 1 Report

Dr. Puddu's group conducted a well-written paper on the putative link between caveolin-1 and VEGF-A in RPE cells. While the concept and work are both outstanding, the authors should address the following comments in order to improve the book.

It is critical to compare the regulation of caveolin-1 and VEGFA by retinal endothelial cells and RPE. Please see the reference below, which revealed an increase in caveolin and transcytosis-related processes following VEGF injection in vivo.

Reference)

Yang JM, Park CS, Kim SH, Noh TW, Kim J-H, Park S, et al. Dll4 Suppresses Transcytosis for Arterial Blood-Retinal Barrier Homeostasis. Circ Res. 2020;126: 767–783. doi:10.1161/CIRCRESAHA.119.316476

Author Response

Thank you for your comments. We appreciated your suggestion and revised the manuscript adding a brief comment in the discussion (lane 270-273) “In this case, downregulating VEGF-A secretion through Cav-1 depletion may be useful in counteracting excessive proangiogenic stimulation. Furthermore, increased levels of VEGF-A in the retina threaten integrity of the blood retinal barrier by inducing transcytosis in endothelial cells (31). All these considerations support the important role of caveolin-1 in maintaining VEGF-A homeostasis.”

Reviewer 2 Report

This short communication describes that silencing of caveolin-1 prevents IGF-1 induced VEGF-A secretion by ARPE-19 cells. The data presented here is clinically relevant; however, the interpretation of the presented data is sometimes not clear to me. Although the manuscript fulfils the criteria of a short communication, I hope that further ex vivo and in vivo experiments will be conducted in the future in order to fully characterize the effect of caveolin-1 on the functions of retinal pigment epithelial cells. Therefore, I do not recommend the current version of this manuscript for publication.

Specific suggestions:

  1. Although cited in the text, Figures 1 c-d are not presented.
  2. The conclusions drawn in lanes 170-171, 206-207, and 264-265 are not supported by the presented results. Perhaps, there are mistakes in the order of samples presented on Figure 3.
  3. Catalog numbers of the primary antibodies should be included in 2.3.
  4. In Figures 1 and 2a, size markers should be provided.
  5. In Figure 2a, the expression of unphosphorylated proteins should be shown.
  6. All abbreviations should be given when first mentioned in the text.
  7. There are minor grammatical or spelling errors in lanes 35, 38, 97, 107, 113, 116, 135, 213, 230, 236, 254, and 264.

Author Response

This short communication describes that silencing of caveolin-1 prevents IGF-1 induced VEGF-A secretion by ARPE-19 cells. The data presented here is clinically relevant; however, the interpretation of the presented data is sometimes not clear to me. Although the manuscript fulfils the criteria of a short communication, I hope that further ex vivo and in vivo experiments will be conducted in the future in order to fully characterize the effect of caveolin-1 on the functions of retinal pigment epithelial cells. Therefore, I do not recommend the current version of this manuscript for publication.

Response: We thank you for your important comments. We revised the manuscript according to your suggestions and we hope that you may accept this version of the manuscript. We are planning other experiments to further characterize the effects of caveolin-1 in RPE cells.

Specific suggestions:

  1. Although cited in the text, Figures 1 c-d are not presented.

Response: We apologize for the mistake. We deleted it in the revised version of the manuscript.

  1. The conclusions drawn in lanes 170-171, 206-207, and 264-265 are not supported by the presented results. Perhaps, there are mistakes in the order of samples presented on Figure 3.

Response: We apologize, there is an error in the chart labels. This mistake makes confusion in the interpretation of the results. We corrected errors and the text: "Under unstimulated conditions mRNA expression of VEGF-A was unaffected by Cav-1 depletion (Fig. 3a), but the rate of VEGF-A secretion was reduced (Fig. 3b)" .

  1. Catalog numbers of the primary antibodies should be included in 2.3.

Response: We added this information in the result section

  1. In Figures 1 and 2a, size markers should be provided.

Response: We added size markers in the figure.

  1. In Figure 2a, the expression of unphosphorylated proteins should be shown.

Response: We revised Figure 2 by adding images of unphosphorylated proteins.

  1. All abbreviations should be given when first mentioned in the text.

Response: We apologize for the mistakes. We added all the abbreviation definitions.

  1. There are minor grammatical or spelling errors in lanes 35, 38, 97, 107, 113, 116, 135, 213, 230, 236, 254, and 264.

Response: We carefully checked the manuscript and corrected all the errors.

Reviewer 3 Report

Review of the manuscript entitled: Caveolin-1 down-regulation reduces VEGF-A secretion induced by IGF- 1 in ARPE-19 cells. The manuscript is well written but some minor corrections should be made.

Double spaces and typing errors in lines: 27, 31, 38, 79, 97, 107, 116, 210

Introduction section is well prepared. In the methodology section, antibody catalog numbers should be provided as they are key reagents.

Lines 99-101, have other reference genes been tested? have the authors checked the stability of actin expression?

Lines 112-117 magnification should be added

Line 135 - something is missing

Author Response

Review of the manuscript entitled: Caveolin-1 down-regulation reduces VEGF-A secretion induced by IGF- 1 in ARPE-19 cells. The manuscript is well written but some minor corrections should be made.

Response: We appreciated your comments and revised the manuscript accordingly.

Double spaces and typing errors in lines: 27, 31, 38, 79, 97, 107, 116, 210

Response: We carefully checked the manuscript and corrected all the errors.

Introduction section is well prepared. In the methodology section, antibody catalog numbers should be provided as they are key reagents.

Response: We added this information in the result section

Lines 99-101, have other reference genes been tested? have the authors checked the stability of actin expression?

Response: We used both β-Actin and GAPDH to normalize gene expression levels. Both the reference genes were stable in all experimental conditions.

Lines 112-117 magnification should be added

Response: We added the magnification (line 117): “Images of the damaged areas were taken at 4X magnification using a microscopic…

Line 135 - something is missing

Response: We apologize for the mistake and corrected the sentence: “All the results obtained in si-CTR ARPE-19 cells are comparable to those obtained in untransfected cells (CTR).

Reviewer 4 Report

Puddu, Sanguineti and Maggi present here various cell biology experiments to investigate the effect of knocking down caveolin-1 on VEGF-A and related pathways. Their work are based on ARPE-19 cells, which are appropriate. The paper builds upon previous findings by them from the 2000s, although the novelty value of the paper is limited by the amount of other work on the topic performed since then.

I have several major technical concerns about the work. Most experiments are based on the transient transfection of siRNAs into these cell lines, with Western blotting (normalised to beta-actin) used as a readout for many experiments. These approaches are fine, but tend to lead to fairly high experimental error, especially if studies are only done n=3 times, as is normally the case here. I therefore do not find it believable that their results should have such tiny error bars (it would be unlikely if they showed standard error of the mean, but is completely unreasonable given that they show standard deviation). This is particularly the case in figure 1, where it appears that all untransfected and control-transfected samples (presumable 3 in total) have yielded utterly identical levels of caveolin-1 and IGF-1R, while the degree of error followed siRNA transfection is also infeasibly small. This is also true, to a lesser degree, in figure 2.

The authors also do not make it clear how they quantify the blots. They use ECL, but it is unclear if this is visualised using some form of imaging system, using X-ray based development of film, etc. Please also clarify what or where “Alliance” software comes from. My worry is this is done by densitometry of an existing image, which would only work if the visualised signal is still within the linear range relative to the amount of protein, and in figure 1 it is not clear if this is the case.

All blots should show size markers, and at least 5 band-widths either side of the signal. Where several blots are grouped together (2a), each should be clearly separated from each other.

Why is beta-actin used as a loading control for the phosphorylation-specific antibodies? Surely general Akt, Erk, etc antibodies should be used, to differentiate between increases in phosphorylation and increases in protein. Additionally, what steps were used to prevent phosphatase activity in the lysates? Where multiple phospho-bands are visible, which are quantified?

Figures 1c and d are referred to in the text and in the figure legend, but are not shown in the paper.

Author Response

Puddu, Sanguineti and Maggi present here various cell biology experiments to investigate the effect of knocking down caveolin-1 on VEGF-A and related pathways. Their work are based on ARPE-19 cells, which are appropriate. The paper builds upon previous findings by them from the 2000s, although the novelty value of the paper is limited by the amount of other work on the topic performed since then.

Response: We thank you for your important comments. We revised the manuscript according to your suggestions. The role of Cav-1 in RPE cells is still controversial, and needs to be further clarified. The novelty of this study consists in demonstrating that Caveolin-1 regulates secretion of VEGF-A in a RPE cell line. Moreover, we provide evidence that depletion of Caveolin-1 prevents IGF-1-induced VEGF-A secretion on ARPE-19 cells, and reduced the migratory potential of their secretory products.

I have several major technical concerns about the work. Most experiments are based on the transient transfection of siRNAs into these cell lines, with Western blotting (normalised to beta-actin) used as a readout for many experiments. These approaches are fine, but tend to lead to fairly high experimental error, especially if studies are only done n=3 times, as is normally the case here. I therefore do not find it believable that their results should have such tiny error bars (it would be unlikely if they showed standard error of the mean, but is completely unreasonable given that they show standard deviation). This is particularly the case in figure 1, where it appears that all untransfected and control-transfected samples (presumable 3 in total) have yielded utterly identical levels of caveolin-1 and IGF-1R, while the degree of error followed siRNA transfection is also infeasibly small. This is also true, to a lesser degree, in figure 2.

Response: Caveolin-1 and IGF-1R are constitutively expressed in RPE cells, therefore it is common to find a little variability, especially using cell lines. Moreover, we used a pool of 3 prevalidated siRNA targeting human caveolin-1 for transient transfection, this method allows to improve efficacy and up to 90% of gene silencing is achieved. The high reproducibility of our results is mainly due to the performance of the reagents that we used. However, if this explanation is not satisfactory, and with Editor agreement, we can provide other images to demonstrate the goodness of our data.

The authors also do not make it clear how they quantify the blots. They use ECL, but it is unclear if this is visualised using some form of imaging system, using X-ray based development of film, etc. Please also clarify what or where “Alliance” software comes from. My worry is this is done by densitometry of an existing image, which would only work if the visualised signal is still within the linear range relative to the amount of protein, and in figure 1 it is not clear if this is the case.

Response: As reported in lines 84-88 bound antibodies were detected using the enhanced chemiluminescence lighting system (LiteAblot EXTEND, EuroClone), according to the manufacturer's instructions. Images were acquired using the instruments Alliance LD2 Uvitec, and the densitometric analysis of the bands of interest was performed using the Alliance software. The UVITEC Alliance LD2 is an advanced imaging system based on proprietary optics and a super-cooled camera. Software is provided with the instruments and is used to quantify the light developed by ECL reaction. The software chooses the exposure time in order to avoid signal saturation.

All blots should show size markers, and at least 5 band-widths either side of the signal. Where several blots are grouped together (2a), each should be clearly separated from each other.

 Response: We revised Figure 2 by adding images of unphosphorylated proteins, and separating each set of blots.

Why is beta-actin used as a loading control for the phosphorylation-specific antibodies? Surely general Akt, Erk, etc antibodies should be used, to differentiate between increases in phosphorylation and increases in protein.

Response: When we started this study we verified whether total levels of IGF1R, Akt and Erk1/2 were affected by Cav-1 depletion and we didn’t find any differences, therefore we normalized phosphorylated protein to β-actin. However, we include the images of unphosphorylated proteins for completeness.

Additionally, what steps were used to prevent phosphatase activity in the lysates? Where multiple phospho-bands are visible, which are quantified?

Response: Cells were lysed in RIPA buffer supplemented with protease and phosphatase inhibitors. We added this information in the Materials and Methods line 74: “ARPE-19 cells were lysed in RIPA buffer supplemented with protease and phosphatase inhibitors.

Figures 1c and d are referred to in the text and in the figure legend, but are not shown in the paper.

Response: We apologize for the mistake. We deleted it in the revised version of the manuscript.

Round 2

Reviewer 1 Report

Authors made adequate changes.

Author Response

Thank you for your comment.

Reviewer 2 Report

The manuscript was substantially improved. Therefore, I suggest the current version of this manuscript for publication in Life.

Author Response

Thank you for your comment.

Reviewer 4 Report

The authors have made notable improvements to the manuscript, and addressed some of my comments, however I still have concerns.

The error bars that the authors show in several figures, and in particular figure 1, still seem highly unlikely. While it is true that the level of endogenous protein in a cell is unlikely to vary significantly, all of their works appears to be based on transient transfection of siRNAs. It seems very unlikely that the level or reproducibility of this transient transfection process would be sufficient to give the data received, even if the efficiency of the siRNA was good. The presence of (more) complete Western blots in the supplementary material is commendable, but I would need to see data from the experimental replicates in order to be convinced (note: this does not need to appear in the final version, if everything is indeed OK).

The level or novelty also still seems to be moderate at best. While it is probably true that no one has shown this particular interaction in this particular cell type, it still does not appear to be a major intellectual advance.

Finally, I would ask the authors to more clearly indicate when they have two blots in the same figure (e.g. by adding borders), and to indicate which band(s) they quantified, in instances when multiple species are visible.

Author Response

The authors have made notable improvements to the manuscript, and addressed some of my comments, however I still have concerns.

The error bars that the authors show in several figures, and in particular figure 1, still seem highly unlikely. While it is true that the level of endogenous protein in a cell is unlikely to vary significantly, all of their works appears to be based on transient transfection of siRNAs. It seems very unlikely that the level or reproducibility of this transient transfection process would be sufficient to give the data received, even if the efficiency of the siRNA was good. The presence of (more) complete Western blots in the supplementary material is commendable, but I would need to see data from the experimental replicates in order to be convinced (note: this does not need to appear in the final version, if everything is indeed OK).

Response: We provide original images of all western blots used for statistical analysis, therefore you can verify the goodness of our results. We hope that your doubts are satisfied.

The level or novelty also still seems to be moderate at best. While it is probably true that no one has shown this particular interaction in this particular cell type, it still does not appear to be a major intellectual advance.

Response: As you wrote, the novelty of this study is that caveolin-1 is an important mediator of IGF-1 action in RPE cells. Indeed, we provide evidence that depletion of Cav-1 prevents IGF-1-induced VEGF-A secretion on ARPE-19 cells and reduces the migratory potential of their secretory products. These results have been never reported from other authors, and could be relevant to design therapeutic strategies to counteract ocular neovascularization.

Finally, I would ask the authors to more clearly indicate when they have two blots in the same figure (e.g. by adding borders), and to indicate which band(s) they quantified, in instances when multiple species are visible.

Response: we added borders to the blots showed in the same figure, and indicated the bands quantified (ERK p42). Section Immunoblotting: “… Densitometric analysis of the bands of interest (p 42 has been used for ERK1/2 quantification) was performed using the Alliance software.”

Round 3

Reviewer 4 Report

I still consider the level of novelty of the paper to be fairly poor. The additional blots are appreciated, and go some way towards addressing my concerns, although the data still seems somewhat suspect. However, if the editor and other reviewers are not concerned, then I will not block publication.

Author Response

Our results have been obtained using validated reagents with adequate lab practice. Obviously, they are true, despite your suspicions.